# Synthesis, Optical, and Geometrical Approaches of New Natural Fatty Acids’ Esters/Schiff Base Liquid Crystals

**DOI:** 10.3390/molecules24234293

**Published:** 2019-11-25

**Authors:** Rua Alnoman, Fares khalid Al-Nazawi, Hoda A. Ahmed, Mohamed Hagar

**Affiliations:** 1College of Sciences, Chemistry Department, Taibah University, Yanbu 30799, Saudi Arabia; rua-b-n@live.co.uk; 2Ibn al-Nafis Secondary School, Yanbu Industrial, Yanbu 46455, Saudi Arabia; Faris-kh16@hotmail.com; 3Faculty of Science, Department of Chemistry, Cairo University, Cairo 12613, Egypt; 4Faculty of Science, Chemistry Department, Alexandria University, Alexandria 21321, Egypt

**Keywords:** liquid crystal, Schiff base, fatty acids, alkenyl terminal group, geometrical analysis, DFT calculations

## Abstract

Schiff base liquid crystals, known as [4-(hexyloxy)phenylimino)methyl]phenyl palmitate (**IA**), [4-(hexyloxy)phenylimino)methyl]phenyl oleate (**IIA**) and [4-(hexyloxy)phenylimino)methyl]phenyl linoleate (**IIIA)**, were synthesized from palmitic, oleic, and linoleic natural fatty acids. The prepared compounds have been investigated for their thermal and optical behavior as well as phase formation using differential scanning calorimetry (DSC) and polarized optical microscopy (POM). Molecular structures of all studied compounds were confirmed via elemental analysis, FT-IR, ^1^H NMR, and ^13^C NMR. Smectic phase is the observed mesophase for all compounds; however, their type and range depend upon the terminal alkanoate chains attached to the phenyl ring. Computational calculations, Density functional theory (DFT), energy difference of the frontier molecular orbital (FMOs), as well as the thermodynamic parameters of different molecular configurations isomers were discussed. It was found that the mesophase behavior and the geometrical characteristics were affected by the degree of unsaturation of fatty terminal chains. Furthermore, the geometrical structure of the CH=N linkage plays an important role in the thermal stability and optical transition temperature.

## 1. Introduction

Optical displays and temperature/humidity sensors are important applications of liquid crystal (LC) in the field of instrumentation [1,2,3,4,5]. The most effective features of these materials that must be considered to be accessible in devices’ applications are the type of the phase, tilt angle, dielectric anisotropy, birefringence anisotropy, switching times, and optical rotation power at convenient working temperatures [6]. Structure-activity relationships could be helpful tools to design a material to achieve the desired properties for device applications [7,8,9,10]. Thus, the choice of the terminal substituents, flexible wings, as well as the mesogenic linkages are important criteria in preparation of thermotropic LCs for proper characteristic applications. Furthermore, the molecular architecture enables some considerable changes in the mesomorphic properties and plays an important role in the formation, type, and stability of the observed mesophase [11,12,13,14,15,16,17,18]. Recently, several imino/ester liquid crystal materials have been investigated and studied in terms of their optical activity [19,20,21,22,23]. The connected CH=N- linkage to rigid phenyl rings provides a stepped core structure and retains the linearity of molecular structure. Thus, it will enhance the stability of the mesophase [24]. Low molar mass LCs comprising just a single mesogenic group [7,17,25,26] exhibiting behavior significantly differs from molecules having two mesogenic units. Thus, attachment of different small compact terminal substituents or alkoxy/alkyl chains will impact the molecular geometry and thermal parameters of the designed LCs [27]. Recently, many researchers studied the design of the molecular deformation architecture influenced the mesophase formation [28,29,30,31], e.g., lattices of free space, fibers in twist-bend nematic phase [32,33], and oligomers [7,34,35,36]. Many of these studies have involved the investigation of the impact of terminal substituents on the temperature of phase transition [37].

It was reported that the mesomorphic properties could be improved by introducing polar substituents of strong dipole moments [38]. The high dipole moment enhances the stability of the molecular packing in the lattice and thus promotes high melting temperatures [39]. On the other hand, the attachment of terminal flexible chains in molecular structure significantly alters their optical and thermal characteristics. High parallel arrangement of LCs observed for long terminals enhances smectic mesophase formation [40]. Therefore, flexible chains, mesogenic cores, and terminal groups play an important role in the designing of new thermotropic liquid crystals [41]. Recently, our research interest focused on the investigation of the molecular and conformational geometry relationship with the thermal and mesomorphic behavior as well as phase transition ranges of the synthesized LC materials [14,17,42,43,44,45,46,47,48,49,50]. We succeeded at connecting the desired properties from experimental data with the estimated parameters by theoretical calculations.

Three new Schiff base LC compounds of two aromatic rings with terminal substituted derivatives (**I**–**III**)**A** have been synthesized (Figure 1). A systematic comparison between the prepared compounds that were attached with a linear n-alkyl chain of palmitic acid and unsaturated alkenyl chains (oleic and linoleic acids) with different degrees of unsaturation is one of the interesting aims of this work. Moreover, the investigation of their mesomorphic behavior and molecular configurations that could exist in principle is of interest. The photophysical properties of different mesogenic cores were studied. Finally, the effect of the fatty chains as well as the geometrical structure of the CH=N on the estimated thermal parameters have been discussed.

The intermediate **A** and title compounds (**I**–**III**)**A** were prepared according to the following Scheme 1:

## 2. Results and Discussion

### 2.1. Characterization of Compounds *(**I**–**III**)**A***

The FT-IR spectra of all investigated compounds (**I**–**III**)**A** were measured by PerkinElmer B25 (PerkinElmer, Inc., Shelton, CT, USA) spectrophotometer. The results of FT-IR showed that the length as well as the degree of unsaturation of the carboxylate chain do not extensively impact the position of absorption bands of the main characteristic functional groups. The absorption peak at 2910–2840 cm^−1^ is assigned for C–H stretching vibration; however, C=O stretching vibration appeared at ~1745 cm^−1^. On the other hand, absorption peak at ~1610 cm^−1^ is the stretching vibration of C=N functional group, C=C stretching vibration appeared at ~1600 cm^−1^. Peaks at ~1470 and ~1200 cm^−1^ were assigned for C–O asymmetric and symmetric stretching vibration, respectively.

In fact, Schiff bases are types of compounds that could exist in two forms, *E* and *Z* isomers; however, the interconversion between them could be achieved either by UV irradiation or thermal heating. From the DSC thermograms, there is only observed one isomer and we could attribute this due to the transformation of the *Z* isomer to the other *E* under thermal heating. Obviously, from the NMR spectroscopy the reaction proceeded to afford a mixture of two geometrical isomers (*E* and *Z*) their ratio 95:5, respectively, (see Appendix A).

### 2.2. Mesomorphic Investigations

The transition temperatures, associated enthalpy, and normalized entropy of transitions, as measured using DSC, are listed in Table 1. Identified phases are obtained by POM for the new synthesized liquid crystal compounds (**I**–**III**)**A**, which are verified by the DSC measurements, and some representative examples are shown in Figure 2. Thermograms of DSC for [4-(hexyloxy)phenylimino)methyl]phenyl linoleate, **IIIA**, are shown in Figure 3 upon heating and cooling scan with rate 10 °C/min. It is clear from the data in Table 1 and phase transitions in Figure 4 that all synthesized compounds are mesomorphic with different mesomorphic stability, and their stability depends on the fatty acid type. Moreover, all compounds are enantiotropic mesomorphic exhibit smectic mesophases. Furthermore, [4-(hexyloxy)phenylimino)methyl]phenyl palmitate, **IA**, is purely smectogenic with narrow smectic C phase range 4.2 °C upon heating. The mesomorphic behavior of **IIA**, [4-(hexyloxy)phenylimino)methyl]phenyl oleate, was affected by the length and the unsaturation of its terminal fatty chain. It enhances SmC range of thermal stability higher than the saturated compound **IA** (9.5 °C). In general, the stability of mesophase is augmented by an increment in the polarizability and/or polarity of the mesogenic portion of whole molecule. In compound **IIIA**, the terminal chain is more lengthened and unsaturated rather than compounds **IA** and **IIA**, so an enhancement in mesophase range was observed. Furthermore, the [4-(hexyloxy)phenylimino)methyl]phenyl linoleate, **IIIA**, is dimorphic exhibiting SmC and SmA mesophases. The molecules tend to orientate in a parallel arrangement with an increase in the length of the terminal groups [40], which in turn enhances the formation of the smectic A phase. Furthermore, the formation of the semectic phase may be attributed to the microphase separation between aromatic cores and the fatty acid chains, which becomes more favorable as the terminal chain length increases [51,52]. It was reported that [17,53] the dipole moment of the mesogenic part of the molecule is affected by the type and stability of the observed mesophase, which is dependent on the attached polar substituent and the steric one that varies according to the size and position of the terminal group. The observed reduction in the clearing temperature for prepared series provides importance of the shape in determining liquid crystal phase behavior. From these observations, it seems that the presence of polarizable portion enhances the intermolecular attractions between molecules, which in turn affects the mesophase stability. It is also shown from Table 1 and Figure 4 that melting transition temperatures irregularly vary with the fatty chain length, which are highly affected by the degree of unsaturation of the chain of the fatty acid. Esters prepared from oleic and linoleic acids produce liquid crystalline compounds with lower melting temperature than the palmitic derivative, which is augmented by the length of the saturated fatty chain. A slightly higher melting temperature of compound **IA** with respect to other compounds was rationalized due to the unsaturation of the terminal fatty acid chain. The end-to-end intermolecular interactions play an important role in determining the smectic-to-isotropic liquid transitions. The development of the smectic molecular order is set by the fact that the parallel attractions become stronger, permitting the arrangement of the layers to occur a lot; consequently, the smectic-to-isotropic liquid transition is enhanced [17]. Association intermolecular interactions that resulted from the conformation changes of the terminal hexyloxy and alkenyl chains influenced the smectic phase observations for all prepared series. Thus, parallel aggregations for the natural fatty acid derivatives are more pronounced than the terminal interactions.

The normalized entropy changes for clearing transitions were estimated for prepared compounds and tabulated in Table 1. As seen from Table 1, the entropy change during phase transitions increases regularly with the increment of degree of unsaturation of terminal fatty chain that is attributed to their high conjugative interactions with the mesogenic portion of the molecules. 

### 2.3. Effect of Different Mesogens and Terminal Groups on the Mesophase Behavior

The effect of the molecular structure changes influences geometrical and mesomorphic properties of liquid crystalline materials. The modeling of a molecule that gives clear information of how the molecular geometry affects mesomorphic transitions still remains very attractive. The comparison between our present compounds and the previously reported related compounds of different mesogens and/or different polar terminal derivatives [13,17,54,55,56,57] were summarized in Table 2. Phase transitions’ comparison of two- and three-ring molecules were made between reported derivatives **IV**, **V**, **VI** with our present series (**I**–**III**)**A**.

Compound **IV** was prepared by the esterification of palmitic acid with lateral hydroxy Schiff bases. It exhibits an enantiotropic SmA phase with a clearing temperature of 111.0 °C. The attachment of a lateral OH group to the existing structure **IA** with short terminal alkoxy chain length (-OCH_3_ instead of -OC_6_H_13_) affects the type and stability of the resulted mesophase. In addition, higher van der Waals forces influence a higher degree of mesophase stability and consequently anisotropy of the molecular polarizability.

Comparison of a previously reported azo derivative **V** and present series (**I**–**III**)**A** reveals that the only difference is a mesogenic core (CH=N instead of N=N) benzene rings with shorter alkoxy chains. It was found that the Schiff base of the present series (**I**–**III**)**A** makes more thermal stability of the present series, where the azo derivative **V** [17,55] exhibits narrow SmA phases.

Compound **VI** showed melting temperature 78.0 °C and enantiotropic N mesophase with higher width (29.0 °C) than series **V**. A longer alkenyl group gave favorable SmA mesomorphism in compounds **V**. These results indicate that the terminal alkoxy groups play an important role in the mesophase behaviors and their range of stabilities. Overall, the introduction of terminal chains at both ends of molecular structure improves the mesophase behavior. Furthermore, alkyl chains are very labile and can easily make multi-conformational changes [58] usually involve contribution from molecular packing effects.

In order to investigate the influence of the chiral carbon atom in a terminal fatty acid chain on the mesomorphic property, a non-photochromic chiral compound was studied [57]. Cholesteric mesophases were induced by its mixing with photochromic chiral azobenzene compounds in a host nematic liquid crystal. In addition, the photo switching between nematic and cholesteric phases was achieved through reversible *trans*/*cis* photoisomerization of the chiral azobenzene molecule through irradiation with UV and visible light. Thus, there is a slight change in the molecular shape of the present Schiff base system (**I**–**III**)**A** capable of changing the photophysical behavior as seen for nematogenic compounds [59].

In order to investigate the effect of an addition of a phenylester component to the present series (**I**–**III**)**A** with dialkoxy terminal chain ends of compounds, the length of the mesogenic core will then extend the clearing transitions of the three-ring compound [13] increased. This additional mesogenic core increases the molecular length. Molecular shapes with an extended core system offer mesophase stability, thus the three-ring system possess SmA and N mesophases with high stability and width mesomorphic range.

Finally, the Schiff base linkage in present compounds (**I**–**III**)**A** offers a stepped core structure with its molecular linearity remaining. Therefore, it provides better stability and inducing formation of mesophase [59]. Therefore, it can be concluded that the Schiff base spacer unit played an important role in decreasing the melting temperature with a suitable range of application.

### 2.4. DFT Calculations

#### 2.4.1. Optimized Geometrical Structures

The theoretical DFT calculations were performed in gas phase by DFT/B3LYP method at the 6–31G (d,p) basis set [60,61]. All optimum compounds are proved to be stable due to the absence of the imaginary frequency. The calculations were carried out for the prepared compounds (**I**–**III**)**A**, and their geometrical isomers, *Z* isomer around the CH=N (**I**–**III**)**B**, Figure 5. The estimated DFT calculations for the thermal parameters, dipole moment, and the polarizability of the compounds under investigations are summarized in Table 3. The results of the theoretical DFT calculations revealed that the *Z* isomers, **B**, for all compounds are less stable than the *E* isomers, **A**. This result could be explained in the terms of the steric effect under which the *Z* isomers are suffering. **A** isomers are more stable than **B** by 7.540776 kcal mol^−1^ for compound **I** derived from palmitic acid, while 7.712086 and 7.767934 kcal mol^−1^ is the energy difference for **II** and **III**, respectively. Moreover, the fatty acid chain type and the degree of unsaturation as well as the geometrical structure have an impact on the dipole moment and the calculated polarizability. The dipole moment and the polarizability of the *E* isomers are higher than that of the *Z* isomers for all types of the fatty acid chains. Recently, our group reported [11,14,16,17,47,48,62] that the length of the alkoxy chains as well as geometrical and conformational structures has a pronounced effect on the competitive lateral interaction and end-to-end aggregation of the chains, where the kind and the degree of the interaction of the liquid crystalline compounds affect the type and the stability of the observed mesophases. The high degree of backing of the chains of the fatty acids could be a good explanation for the enhancement of the smectic C mesophases; as the unsaturation increasing the lower backing of the chains is observed, the less ordered smectic A mesophase enhanced. The high degree of interaction of the linear shape of palmitic acid derivative **IA** could permit the strongest lateral interaction to afford a smectic C mesophase with the highest mesophase stability, ***T****_Cr-SmC_ =* 93.5 °C. However, as the unsaturation of the fatty acid part increases, the linearity of the chains decreases and, consequently, lateral interaction between the molecules decreases to afford smectic A mesophase for linoleic acid derivative **IIIA,** T_SmC − SmA_ = 74.0 °C and T_SmA − I_ = 83.8 °C.

#### 2.4.2. Frontier Molecular Orbitals and Molecular Electrostatic Potential

Figure 6 shows the estimated ground state isodensity surface plots for the FMOs HOMO (highest occupied molecular orbital) and LUMO (lowest unoccupied molecular orbital)) as well as their energy difference (**ΔE**) of the compounds under investigation, **IA**, **IB**, **IIA**, **IIB**, **IIIA**, and **IIIB**. As shown from Table 4, FMO energy gap and the global softness (**S**) were not significantly affected by the type of the fatty acid chain or its geometrical isomerism. This could be explained in terms of the unchanged electronic nature with the type of the alkenyl fatty acid chain. However, the CH=N geometry has a pronounced effect on the energy difference between the FMOs as well as the global softness (**S**). The results expected that the *E* isomers **A** are softer than their *Z* isomers **B**. This data could be attributed to the proper structure of the *Z* isomers for more conjugation of the π-bonds rather than the *E* isomers.

#### 2.4.3. Photophysical Behavior of the Investigated Compounds and Their Corresponding Azo Analogues

Absorption spectra of present Schiff base series **I**–**III** and their corresponding previously prepared azo derivatives **V**(**i**–**iii**) were performed in dichloromethane solutions at 25 °C, and their absorption bands were summarized in Figure 7. As can be seen from Figure 7, the present compounds with -N=CH- linkages (**I**–**III**) exhibit two absorption bands at 274 nm (ε ~ 3.7 × 10^4^ L. mol^−1^) and 341 nm (ε ~ 3.2 × 10^4^ L. mol^−1^) while the azobenzene derivative (**V**(**i**–**iii**)) showed only one electronic transition absorbance peak at 352 nm. These observed peaks may be attributed to the π–π* transition involved in the π-electronic system throughout the whole mesogenic portion, with a suitable charge transfer (CT) property [63,64,65]. The photoisomerization for present compounds has confirmed and occurred in the range of 300–400 nm. The absorption peaks were observed at 274 and 341 nm, corresponding to π–π* and n–π* as these suggesting that two isomers obtained in Schiff base derivatives **I**–**III**. *E*–*Z* isomerization that could be occurred with a gradual decrease in absorption band from 274 or 352 nm due to π–π* transition of *trans* (*E*) isomer in azo group (N=N) or imino group (CH=N), respectively, Figure 8. In addition, a smaller absorption band appeared around 505 nm for (N=N) and 341 for (CH=N), appropriate to n–π* transition of *cis* (*Z*) isomer that could be gradually changed by photo illumination. The transformed *cis* isomer (*Z*) is converted to *trans* form (*E*) due to the -CH=N- group enhancement the electron delocalization on the whole molecule. Moreover, the investigation of such effects on the stability of the obtained isomers could be an important point for future perspectives either theoretically or experimentally.

It is noteworthy that there is no significant effect on the absorption peaks position either for the imino derivatives **I**–**III** or their azo analogues **V**(**i**–**iii**). This result could be explained in terms of the estimated data by DFT calculations, Table 4. The change of the alkyl of the alkenyl groups do not affect the polarity of the molecules where these groups do not reduce the HOMO–LUMO energy gaps of the molecules as shown in Figure 6**.** Moreover, the terminal alkanoate or alkenoate groups has no effect on the energy difference between the frontier molecular orbitals (FMOs), ΔE ≈ 0.1427 a.u. for imino derivatives and 0.1362 a.u. for azo one. Obviously, the higher energy difference in case of the imino derivative **I**–**III** makes them less polarizable than their azo derivatives **V**(**i**–**iii**)—that of a lower energy difference, Table 4 and Figure 7 and Figure 8.

## 3. Experimental

### 3.1. Synthesis

#### 3.1.1. Synthesis of 4-(Hexyloxy)phenylimino)phenol **A**

Equimolar amounts of the 4−hydroxybenzaldehyde (0.5 g, 4.1 mmol) and 4-hexyloxyaniline (0.79 g, 4.1 mmol) were dissolved in ethanol (10 mL) and refluxed for two hours. The mixture was cooled to room temperature and filtered. The obtained solid was washed with cold ethanol and recrystallized twice from hot ethanol to give pure compounds, as indicated by TLC analysis

#### 3.1.2. Synthesis of [4-(Hexyloxy)phenylimino)methyl]phenyl carboxylate (**I**–**III**)**A**

Equimolar equivalents of fatty acid (4.1 mmol) and 4-(hexyloxy)phenylimino)phenol **A** (1.21 g, 4.1 mmol) were dissolved in 25 mL dry methylene chloride. Furthermore, 4-dimethylaminopyridine (DMAP, few crystals), as catalyst, and 0.02 mol of *N*,*N′*-dicyclohexylcarbodiimide (DCC) were added to the reaction mixture. At room temperature, the reaction mixture was stirred for 72 h. The obtained solid byproduct was filtered. The remained filtrate was evaporated, and the solid obtained was twice recrystallized from ethanol.

##### 4-((*E*)-9-*cis*-[4-(Hexyloxy)phenylimino)methyl]phenyl hexadec-9-enoate **IA**

Yield: 93.7%; m.p. 93.5 °C, FTIR (ύ, cm^−1^): 2918–2859 (CH_2_ stretching), 1744 (C=O), 1615 (C=N), 1601 (C=C), 1471 (C−O_Asym_), 1230 (C–O _Sym_). ^1^H NMR (300 MHz, CDCl_3_) δ 8.39 (s, 1H), 8.00 (d, *J* = 8.8, Hz, 2H), 7.30 (d, *J* = 6.9, Hz, 2H), 7.13 (d, *J* = 8.9, Hz, 2H), 6.96 (d, *J* = 7.0, Hz, 2H), 3.90 (t, *J* = 6.6 Hz, 2 H), 2.50 (t, *J* = 7.5 Hz, 2H), 1.82–1.60 (m, 4H), 1.58–1.05 (m, 30H), 0.97–0.69 (m, 6H). ^13^C NMR (75 MHz, CDCl_3_) δ 172.16, 156.87, 134.12, 133.98, 129.84, 129.77, 122.53, 122.05, 121.98, 114.99, 68.31, 34.45, 31.94, 31.61, 29.70, 29.67, 29.61, 29.47, 29.37, 29.28, 29.12, 25.74, 24.86, 22.68, 22.62, 14.13, 14.05. Elemental analyses: Found (Calc.): C, 78.43 (78.46); H, 9.94 (9.97); N, 2.60 (2.61).

##### 4-((*E*)-9-*cis*-[4-(Hexyloxy)phenylimino)methyl]phenyl octadec-9-enoate **IIA**

Yield: 91.0%; m.p. 68.0 °C, FTIR (ύ, cm^−1^): 2919–2858 (CH_2_ stretching), 1749 (C=O), 1611 (C=N), 1587 (C=C), 1470 (C−O_Asym_), 1210 (C–O_Sym_). ^1^H NMR (300 MHz, CDCl_3_): *δ*/ppm: ^1^H NMR (300 MHz, CDCl_3_) δ 8.39 (s, 1H), 7.99 (d, *J* = 8.7, Hz, 2H), 7.45 (d, *J* = 6.9 Hz, 2H), 7.12 (d, *J* = 6.9, Hz, 2H), 6.98 (d, *J* = 8.8, Hz, 2H), 5.46–5.11 (m, 2H), 3.90 (t, *J* = 6.6 Hz, 2H), 2.50 (t, *J* = 7.5 Hz, 2H), 2.08–0.93 (m, 32H), 0.92–0.70 (m, 6H). ^13^C NMR (75 MHz, CDCl_3_) δ 171.82, 157.02, 134.01, 132.87, 131.19, 130.01, 129.81, 129.71, 122.16, 121.97, 114.99, 68.31, 49.15, 33.97, 29.69, 29.33, 29.28, 29.17, 29.10 25.74, 24.95, 22.62, 14.18, 14.05. Elemental analyses: Found (Calc.): C, 79.07 (79.10); H, 9.85 (9.87); N, 2.48 (2.49).

##### 4-((*E*)-9, 12-*cis*-[4-(Hexyloxy)phenylimino)methyl]phenyloctadeca-9,12-dienoate **IIIA**

Yield: 90.1%; m.p. 61.0 °C, FTIR (ύ, cm^−1^): 2915–2860 (CH_2_ stretching), 1748 (C=O), 1610 (C=N), 1589 (C=C), 1470 (C−O_Asym_), 1209 (C–O_Sym_).). ^1^H NMR (300 MHz, CDCl_3_): *δ*/ppm: ^1^H NMR (300 MHz, CDCl_3_) δ 8.39 (s, 1H), 7.84 (d, *J* = 8.8, Hz, 2H), 7.18 (d, *J* = 6.9 Hz, 2H), 7.14 (d, *J* = 6.8, Hz, 2H), 6.81 (d, *J* = 8.9, Hz, 2H), 5.55–5.24 (m, 4H), 3.90 (t, *J* = 6.6 Hz, 2H), 2.71 (t, *J* = 6.9 Hz, 2H), 2.50 (t, *J* = 6.8 Hz, 2H), 2.21–1.07 (m, 28H), 1.01–0.30 (m, 6H). ^13^C NMR (75 MHz, CDCl_3_) δ 171.94, 156.89, 130.24, 130.06, 130.01 129.72, 128.10, 127.89, 127.82, 122.17, 121.98, 115.00, 68.31, 49.05, 33.97, 31.61, 29.35, 29.10, 27.20, 25.74, 24.95, 24.89, 22.62, 14.09, 14.02. Elemental analyses: Found (Calc.): C, 79.37 (79.38); H, 9.50 (9.54); N, 2.46 (2.50).

## 4. Conclusions

Optical transition behavior and isomeric geometrical calculations of new non-symmetric Schiff base natural fatty acid derivatives were investigated. Molecular structures were elucidated via elemental analysis, FT-IR, ^1^H NMR, ^13^C NMR, and spectroscopy. Mesomorphic and optical characterizations were investigated by DSC, POM, and UV spectroscopy. Computational and geometrical parameters were studied using Gaussian 09 software (DFT/B3LYP methods using 6–31G (d,p) basis set). All prepared palmitic, oleic, and linoleic natural fatty acids-based LCs are exhibiting smectogenic mesophases. The type and stability of observed mesophases depend on the length and conformation of the terminal alkenyl fatty acid chains. Computational calculations of different structure configuration isomers for each compound revealed that the geometrical and thermal parameters characteristics are influenced by the degree of unsaturation of fatty alkenyl terminal chains. Furthermore, the present Schiff base CH=N linkage molecular configurations offer high thermal and different optical transition phases than their corresponding azobenzene analogues.

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
