# Peer review of "Synthesis, Optical, and Geometrical Approaches of New Natural Fatty Acids’ Esters/Schiff Base Liquid Crystals"

_molecules, 2019, doi:10.3390/molecules24234293_

Round 1
Reviewer 1 Report
In their manuscript Alnoman et al. describe synthesis and analysis of several Schiff base liquid-crystalline compounds. This is a topic of constant interest both for fundamental physical chemistry and for various applications.
The manuscript has considerable scientific merit, and, as far as I can see, the reported results are new and original. The synthesized compounds are analyzed with the aid of different methods and their behavior is characterized in detail. Unfortunately I must say that the manuscript is not clearly written. Although English is not my native language, I have an impression that the manuscript needs rather extensive editing. The meaning of some sentences is unclear or seems to be very trivial.
The conclusions are also not very clear. In particular, in lines 358-359 the authors write: “…it could be concluded that the possessing of photoisomerization phenomena depends on type of mesogens”. This claim is rather trivial, although it might reflect some of the authors’ findings. Next, the authors write: “Finally, the CH=N linkage molecular configurations offer new thermal and optical transitions”. Probably the authors mean “new” transitions with respect to those observed in previously investigated compounds without the CH=N bond; in this case it should be explicitly stated, because the transitions are not new in the general sense. The abovementioned sentences should be reformulated.
In summary, I suggest a revision of the manuscript and encourage the authors to resubmit the amended version to Molecules.
Author Response
I would like first to thank you for his valuable and accurate comments that helped us to revise the manuscript more thoroughly. All his suggestions have been considered in the revised manuscript in a red color.
The manuscript has considerable scientific merit, and, as far as I can see, the reported results are new and original. The synthesized compounds are analyzed with the aid of different methods and their behavior is characterized in detail. Unfortunately I must say that the manuscript is not clearly written. Although English is not my native language, I have an impression that the manuscript needs rather extensive editing. The meaning of some sentences is unclear or seems to be very trivial.
English language has been checked.
The conclusions are also not very clear. In particular, in lines 358-359 the authors write: “…it could be concluded that the possessing of photoisomerization phenomena depends on type of mesogens”. This claim is rather trivial, although it might reflect some of the authors’ findings. Next, the authors write: “Finally, the CH=N linkage molecular configurations offer new thermal and optical transitions”. Probably the authors mean “new” transitions with respect to those observed in previously investigated compounds without the CH=N bond; in this case it should be explicitly stated, because the transitions are not new in the general sense. The abovementioned sentences should be reformulated.
The conclusion has been cleared according to your advise.
Reviewer 2 Report
This manuscript shows the synthesis, phase-transition and optical properties for three of Schiff base liquid crystals containing long alkanoate flexible unit, which are derived from natural fatty acids, e.g., palmitic, oleic, and linoleic acid. Very recently, authors reported the impact of the alkoxy chain length on the mesophase behavior for some Schiff base/ester liquid crystals from both of the experimental and the theoretical aspects. This reviewer can understand that the research has been done by a course of the authors’ research project, but the novelty cannot be found in this manuscript. Although the three of molecules derived from the corresponding natural fatty acid may be novel, but the phase-transition phenomenon is just usual, i.e., incorporation of cis-configuration into the molecular structure decrease phase-transition temperature. Therefore, this reviewer cannot accept this manuscript to publish to Molecules as it stands now. However, this reviewer would reconsider after suitable modifications, e.g., addition of new experimental results and systematic discussions, etc., including the following points.
Major points:
[1] Introduction part: this reviewer cannot understand clearly the aim of your research described in this manuscript, therefore, that authors should rewrite concisely the introduction part.
[2] Line 93 (Experimental part): Authors described as “from the NMR spectroscopy, the reaction proceeded to afford a mixture of two geometrical isomers (E and Z) their ratio 95:5%, respectively, (see supplementary data).” In general, the syn isomer of the imine derivatives are thermally unstable, which would be less formed. Authors may assign as Z isomer at the small 1H signals neighboring large ones in the NMR spectrum. Is the assignment correct?
If correct, is the phase-transition behaviors, i.e., phase sequence and temperature, reliable? Syn-isomers should influence on the phase transition even in only 5%. At least, authors should add any comment or description at the effect of syn-isomer on the phase transition behavior in the manuscript.
[3] In the section 2.2. Mesomorphic investigations: Authors assigned the mesomorphic texture for IIA and IIIA as smectic A as well as smectic C, which may be assigned from POM images. This reviewer suggests to make XRD experiment, which would more precisely assigned the smectic phases.
[4] Line 185-187: The description about the compound IX does not match with the chemical structure for IX. Author should correct the description or chemical structure for IX.
[5] In the section 2.4. Optimized geometrical structures: In the section, authors mentioned about the present structures (I-III)A as well as the (II-III)C with E-configured alkene moiety in the oleic and linoleic acid derivatives. This reviewer cannot understand why authors add the results on the (II-III)C at all. If authors discuss about the alkene-configuration, not only the theoretical results but also experimental ones should be added and comprehensively discussed; this reviewer suggests to remove from the manuscript if author does not add the results on such compounds.
[6] In the section 2.4.3. Photophysical behavior of the …: Authors disclose the absorption spectra for the present compounds, I-III, as well as the previously reported diazo compounds as a comparison. This reviewer is wondering why the photophysical data between C=N and N=N compounds are compared, because such photophysical behavior for C=N and N=N compounds were deeply investigated. This reviewer feels that the photophysical data for N=N compounds is not necessary in this manuscript. If authors feel such data should be needed in this manuscript, please explain the necessity in this report.
[7] Line 263: Authors mentioned as “a smaller absorption band appeared around 505 nm for (N=N) and 341 for (CH=N), appropriate to n-π* transition of cis-(Z) isomer …”. Please add the molar extinction coefficient (ε) of the signals at 274 nm and 341 nm. If the ε values are as high as the order of 10,000, the absorption signal at 341 nm is not n-π* transition because the n-π* transition is generally forbidden transition. In contrast, the epsilon values are less than 1,000, the absorption signal at 274 nm cannot be assigned as π-π* transition.
[8] In the section 2.4.4. Molecular electrostatic potentials (MEP): This reviewer cannot understand what authors discuss from the MEP distributions. In lines 295-297, although authors mentioned that “the presence of the double bonds of oleic and linoleic acids chains decreases the chance of high backing of the chains to affect the type and the stability of the mesophase formation and it become effective to enhance the smectic A mesophase in case of the linoleic acid chain IIIA with the two double bonds of the least linearity.”, the results can be sufficiently led from other calculation results, not the MEP result. This reviewer suggests to remove this section of MEP if there is no more promising understanding.
Minor points:
[9] Line 20: C13 should be “13C”.
[10] Line 49: “of the” is duplicate. Please remove one.
[11] Figure in pages 3-4: compounds IIC and IIIC should be removed because such compounds will make most readers confused.
[12] Table 2 in page 8: the compound structure at the ester unit should be “OCOC15H31”, not “OOC15H31”.
[13] Table 2 in page 9-10: why are phase transitions for VII, VIII, and IX brank? If no, the columns for compounds VII, VIII, and IX should be removed.
[14] Experimental in page 16: Please add the assignment of the syn-isomers of I-IIIA.
[15] Experimental in page 16: Please add the 13C signals obtained from 13C NMR and the spectra should be added in the supplementary data.
[16] In the reference sections: the format for the references should be consolidated according to the manuscript template.
[17] Page numbers for some references should be wrong, ex. Ref.14, 16, 18, 48. Please check all references again and add the correct page numbers.
[18] lines 391 and 395: the references 11 and 14 should be same. Please remove one.
[19] In Supplementary data: Please add the experimental details for UV-Vis absorption measurements, including instrument set-up, solvent, concentration, etc.
Author Response
I would like first to thank you for your valuable and accurate comments that helped us to revise the manuscript more thoroughly. All your suggestions have been considered in the revised manuscript in a red color.
[1] Introduction part: this reviewer cannot understand clearly the aim of your research described in this manuscript, therefore, that authors should rewrite concisely the introduction part.
The introduction part has been cleared according to your advice.
[2] Line 93 (Experimental part): Authors described as “from the NMR spectroscopy, the reaction proceeded to afford a mixture of two geometrical isomers (E and Z) their ratio 95:5%, respectively, (see supplementary data).” In general, the syn isomer of the imine derivatives are thermally unstable, which would be less formed. Authors may assign as Z isomer at the small 1H signals neighboring large ones in the NMR spectrum. Is the assignment correct?
If correct, is the phase-transition behaviors, i.e., phase sequence and temperature, reliable? Syn-isomers should influence on the phase transition even in only 5%. At least, authors should add any comment or description at the effect of syn-isomer on the phase transition behavior in the manuscript.
It is known that Schiff bases are type of compounds that could exist in two forms, E and Z isomers; however, the interconversion between them could be achieved either by UV irradiation or thermal heating. From the DSC curve there is only one isomer and we could attribute this due to the transformation of the Z isomer to the other E under thermal heating and It is mentioned in the revised manuscript.
[3] In the section 2.2. Mesomorphic investigations: Authors assigned the mesomorphic texture for IIA and IIIA as smectic A as well as smectic C, which may be assigned from POM images. This reviewer suggests to make XRD experiment, which would more precisely assigned the smectic phases.
In fact we have compared the obtain textures from the optical microscope with the previously reported; unfortunately, we do not have the access for doing the XRD measurements.
[4] Line 185-187: The description about the compound IX does not match with the chemical structure for IX. Author should correct the description or chemical structure for IX.
The description of IX was corrected.
[5] In the section 2.4. Optimized geometrical structures: In the section, authors mentioned about the present structures (I-III)A as well as the (II-III)C with E-configured alkene moiety in the oleic and linoleic acid derivatives. This reviewer cannot understand why authors add the results on the (II-III)C at all. If authors discuss about the alkene-configuration, not only the theoretical results but also experimental ones should be added and comprehensively discussed; this reviewer suggests to remove from the manuscript if author does not add the results on such compounds.
In fact we have added the comparative discussion of the (I-III)A as well as the (II-III)C with the experimentally prepared compounds to show the effect of the isomerization of the CH=N and the C=C groups on the thermal and the optical parameters which could be important to the reader. However, we have removed it according to your advice.
[6] In the section 2.4.3. Photophysical behavior of the …: Authors disclose the absorption spectra for the present compounds, I-III, as well as the previously reported diazo compounds as a comparison. This reviewer is wondering why the photophysical data between C=N and N=N compounds are compared, because such photophysical behavior for C=N and N=N compounds were deeply investigated. This reviewer feels that the photophysical data for N=N compounds is not necessary in this manuscript. If authors feel such data should be needed in this manuscript, please explain the necessity in this report.
Photophysical data between C=N and N=N compounds are compared to show the effect of the change of the mesogenic core as well as the terminal substituent on the optical and thermal behaviors of compounds. In addition, the shape of molecular structure is an important parameter that affects on the photophysical behaviors. Further, this kind of comparison is good for the liquid crystal readers.
[7] Line 263: Authors mentioned as “a smaller absorption band appeared around 505 nm for (N=N) and 341 for (CH=N), appropriate to n-π* transition of cis-(Z) isomer …”. Please add the molar extinction coefficient (ε) of the signals at 274 nm and 341 nm. If the ε values are as high as the order of 10,000, the absorption signal at 341 nm is not n-π* transition because the n-π* transition is generally forbidden transition. In contrast, the epsilon values are less than 1,000, the absorption signal at 274 nm cannot be assigned as π-π* transition.
The molar extinction coefficient (ε) has been added.
[8] In the section 2.4.4. Molecular electrostatic potentials (MEP): This reviewer cannot understand what authors discuss from the MEP distributions. In lines 295-297, although authors mentioned that “the presence of the double bonds of oleic and linoleic acids chains decreases the chance of high backing of the chains to affect the type and the stability of the mesophase formation and it become effective to enhance the smectic A mesophase in case of the linoleic acid chain IIIA with the two double bonds of the least linearity.”, the results can be sufficiently led from other calculation results, not the MEP result. This reviewer suggests to remove this section of MEP if there is no more promising understanding.
It has been removed.
Minor points:
[9] Line 20: C13 should be “13C”.
It has been corrected
[10] Line 49: “of the” is duplicate. Please remove one.
It has been addressed
[11] Figure in pages 3-4: compounds IIC and IIIC should be removed because such compounds will make most readers confused.
It has been removed
[12] Table 2 in page 8: the compound structure at the ester unit should be “OCOC15H31”, not “OOC15H31”.
It has been addressed
[13] Table 2 in page 9-10: why are phase transitions for VII, VIII, and IX brank? If no, the columns for compounds VII, VIII, and IX should be removed.
It has been addressed
[14] Experimental in page 16: Please add the assignment of the syn-isomers of I-IIIA.
It has been added
[15] Experimental in page 16: Please add the 13C signals obtained from 13C NMR and the spectra should be added in the supplementary data.
It has been addressed.
[16] In the reference sections: the format for the references should be consolidated according to the manuscript template.
It has been addressed.
[17] Page numbers for some references should be wrong, ex. Ref.14, 16, 18, 48. Please check all references again and add the correct page numbers.
It has been addressed.
[18] lines 391 and 395: the references 11 and 14 should be same. Please remove one.
It has been removed
[19] In Supplementary data: Please add the experimental details for UV-Vis absorption measurements, including instrument set-up, solvent, concentration, etc.
It has been added
Reviewer 3 Report
The manuscript #molecules-631968 entitled “Synthesis, optical and geometrical approaches of new natural fatty acids esters/Schiff base liquid crystals” reports the synthesis, mesomorphic behavior and computational study on three Schiff base compounds based on palmitic, oleic and linoleic natural fatty acids. The synthetized compounds were characterized by NMR and FT-IR methods. English language and style require extensive editing. In section 2.2. lines 127-128 the authors state “The high melting temperature of compound IA was rationalized due to the unsaturation of the terminal fatty acid chain”, while both chains of Ia are saturated. The information provided is confusing as to what was actually performed. The authors provided the mesomorphic investigations of compounds (I –III)A and which are then compared with the ones of different reported mesogens. Thus, the authors state at line 166 “comparing with our previously reported azo derivative V and present series (I-III)A …”. In table 2 for compound V the reference [54] is mentioned, which seems is has no connection with the authors of the manuscript but neither with the structure from the cited paper [McCaffrey, M. T.; Castellano, J. A., Liquid Crystals VII. The Mesomorphic Behavior of Homologous: p-494 Alkoxy-p'-Acyloxyazoxybenzenes. Molecular Crystals and Liquid Crystals 1972, 18, (3-4), 209-225]. Moreover, at line 169, is stated a different reference, which is connected with the authors “…azo derivative V [18] exhibiting narrow SmA phases”. In table 1 only the temperature transitions on heating are given but not on cooling. The light microscope images have not good resolution and XRD patterns to attribute the SmC liquid crystalline phases need to be included. In table 2 the structure for compound IV should be reconsidered and overall phase transitions as well; for example Cr – Sm – I. The nomenclature for geometrical isomers in 2.4.1 section should be unitary: E/Z or cis/trans. The last part in conclusions should be reconsidered.
The manuscript should be re-styled and condensed, the terminology should be uniform and appropriate, and more explicit evaluations of the products should be provided.
I recommend major revision.
Author Response
I would like first to thank you for your valuable and accurate comments that helped us to revise the manuscript more thoroughly. All your suggestions have been considered in the revised manuscript in a red color.
In section 2.2. lines 127-128 the authors state “The high melting temperature of compound IA was rationalized due to the unsaturation of the terminal fatty acid chain”, while both chains of Ia are saturated. The information provided is confusing as to what was actually performed.
We mean that the low melting point of IIA and IIIA with respect to IA with the palmetate fatty chain moiety (It has been cleared in revised manuscript).
The authors provided the mesomorphic investigations of compounds (I –III)A and which are then compared with the ones of different reported mesogens. Thus, the authors state at line 166 “comparing with our previously reported azo derivative V and present series (I-III)A …”. In table 2 for compound V the reference [54] is mentioned, which seems is has no connection with the authors of the manuscript but neither with the structure from the cited paper [McCaffrey, M. T.; Castellano, J. A., Liquid Crystals VII. The Mesomorphic Behavior of Homologous: p-494 Alkoxy-p'-Acyloxyazoxybenzenes.Molecular Crystals and Liquid Crystals 1972, 18, (3-4), 209-225].
Moreover, at line 169, is stated a different reference, which is connected with the authors “…azo derivative V [18] exhibiting narrow SmA phases”.
Compound V was reported also by Hagar et.al , 2019 (comparison part has been checked and corrected).
In table 1 only the temperature transitions on heating are given but not on cooling. The light microscope images have not good resolution and XRD patterns to attribute the SmC liquid crystalline phases need to be included.
In fact we have compared the obtain textures from the optical microscope with the previously reported; unfortunately, we do not have the access for doing the XRD measurements.
In table 2 the structure for compound IV should be reconsidered and overall phase transitions as well; for example Cr – Sm – I.
It has been included.
The nomenclature for geometrical isomers in 2.4.1 section should be unitary: E/Z or cis/trans. The last part in conclusions should be reconsidered.
We have remove the cis and trans isomers of the C=C of the fatty acid as the advice of the other reviewer.
Round 2
Reviewer 1 Report
The authors improved the presentation of their results, in particular they worked on the English of their manuscript. However, in the revised version many sentences still contain errors. To give just a few examples:
"Nowadays, several imino/ ester liquid crystal materials have been investigated and studied their optical activity" (lines 43-44)
"Thus it will enhancement the stability of the mesophase" (lines 47-48)
"High parallel arrangement of molecules observed for long terminals [40], that also enhancement the smectic mesophases." (lines 66-67)
"all synthesized series is mesomorphic with different mesomorphic stability depends on the fatty chain length" (lines 124-125)
"it remain molecular linearity" (line 216)
"E–Z isomerization that could be occur with a gradual decrease in absorption band from 274 or 352 nm" (line 288-289)
I suppose the scientific merit of the manuscript deserves publication, but further editing and improvement of the presentation is necessary.
Author Response
I would like first to thank you for your valuable and accurate comments that helped us to revise the manuscript more thoroughly. All your suggestions have been considered in the revised manuscript in a red color
"Nowadays, several imino/ ester liquid crystal materials have been investigated and studied their optical activity" (lines 43-44)
It has been corrected
"Thus it will enhancement the stability of the mesophase" (lines 47-48)
It has been corrected
"High parallel arrangement of molecules observed for long terminals [40], that also enhancement the smectic mesophases." (lines 66-67)
It has been corrected
"all synthesized series is mesomorphic with different mesomorphic stability depends on the fatty chain length" (lines 124-125)
It has been corrected
"it remain molecular linearity" (line 216)
It has been corrected
"E–Z isomerization that could be occur with a gradual decrease in absorption band from 274 or 352 nm" (line 288-289)
Sincerely
Hagar
It has been corrected
Reviewer 2 Report
It seems that the revised manuscript is suitably revised by the authors. Therefore, this reviewer would accept this manuscript to publish to Molecules after minor corrections as follows:
A paragraph lines 209–214 (page 8) should be deleted because the authors delete three columns for VII, VIII, and IX in Table 2 in the revised manuscript. Line 281 in page 14 and Line 282 in page 15: the authors added molar extinction coefficient (ε), but the values are too large. From Lambert-Beer law, A = ε x c x l, in where A: absorbance, c: concentration (mol L–1), and l = optical path length (cm); calculating from the concentration of the solution (2.1x10–5 mol L–1) mentioned in the Supplementary data, the e should be 104 Isn’t the e value 3.7x104 L mol–1 cm–1 for the absorption band at 274 nm? Please check it again and correct adequately. This reviewer found some typos even in the revised manuscript. The manuscript should be checked carefully by using a proofreading service.
Author Response
I would like first to thank you for your valuable and accurate comments that helped us to revise the manuscript more thoroughly. All your suggestions have been considered in the revised manuscript in a red color
A paragraph lines 209–214 (page 8) should be deleted because the authors delete three columns for VII, VIII, and IX in Table 2 in the revised manuscript.
It have been rephrased and cleared according the advice of Reviewer 3
Line 281 in page 14 and Line 282 in page 15: the authors added molar extinction coefficient (ε), but the values are too large. From Lambert-Beer law, A = ε x c x l, in where A: absorbance, c: concentration (mol L–1), and l = optical path length (cm); calculating from the concentration of the solution (2.1x10–5 mol L–1) mentioned in the Supplementary data, the e should be 104 Isn’t the e value 3.7x104 L mol–1 cm–1for the absorption band at 274 nm? Please check it again and correct adequately.
It has been corrected
Sincerely
Hagar
Reviewer 3 Report
Although the revised manuscript has been improved, I accept this manuscript to publish after minor corrections as follows:
Considering the poor resolution of POM images and authors statement “do not have the access for doing the XRD measurements”, I suggest to reconsider the assignment of the mesomorphic texture for I-IIIA compounds as smectic C – e.g only smectic or smectic x. In the section 2.3, line 209-211: some corrections were made, but the phrase doesn’t have sense: ” An incorporation of a phenylester core to our present series (I-III)A with dialkoxy terminal chains, the extended mesogenic core will increase then the stability and clearing transition of the three rings compound IX [13] increased”. More than that, it seems that compounds VII-IX have been erased from table 2, but reference is still made to them, line 201-208. In the section 4, 13C-NMR should be added at spectroscopy investigations. The manuscript still requires extensive editing.
Author Response
I would like first to thank you for your valuable and accurate comments that helped us to revise the manuscript more thoroughly. All your suggestions have been considered in the revised manuscript in a red color.
Considering the poor resolution of POM images and authors statement “do not have the access for doing the XRD measurements”, I suggest to reconsider the assignment of the mesomorphic texture for I-IIIA compounds as smectic C – e.g only smectic or smectic x.
POM textures were checked carefully and agreement with the reported textures
In the section 2.3, line 209-211: some corrections were made, but the phrase doesn’t have sense: ” An incorporation of a phenylester core to our present series (I-III)A with dialkoxy terminal chains, the extended mesogenic core will increase then the stability and clearing transition of the three rings compound IX [13] increased”. More than that, it seems that compounds VII-IX have been erased from table 2, but reference is still made to them, line 201-208.
The section was rephrased and cleared.
In the section 4, 13C-NMR should be added at spectroscopy investigations.
13C-NMR has been added according to our advice.
Sincerely
Hagar